# CpG Methylation Altered the Stability and Structure of the i-Motifs Located in the CpG Islands

**DOI:** 10.3390/ijms23126467

**Published:** 2022-06-09

**Authors:** Daiki Oshikawa, Shintaro Inaba, Yudai Kitagawa, Kaori Tsukakoshi, Kazunori Ikebukuro

**Affiliations:** 1Graduate School of Management of Technology, Tokyo University of Agriculture and Technology, 2-24-16 Naka-cho, Koganei, Tokyo 184-8588, Japan; daiki.oshikawa@gmail.com; 2Department of Biotechnology and Life Science, Tokyo University of Agriculture and Technology, 2-24-16 Naka-cho, Koganei, Tokyo 184-8588, Japan; s218387v@st.go.tuat.ac.jp (S.I.); s214254w@st.go.tuat.ac.jp (Y.K.); k-tsuka@cc.tuat.ac.jp (K.T.)

**Keywords:** CpG methylation, epigenetics, CpG islands, i-motif, G-quadruplex, pH dependance, structural change, thermal stability

## Abstract

Cytosine methylation within the 5′-C-phosphate-G-3′ sequence of nucleotides (called CpG methylation) is a well-known epigenetic modification of genomic DNA that plays an important role in gene expression and development. CpG methylation is likely to be altered in the CpG islands. CpG islands are rich in cytosine, forming a structure called the i-motif via cytosine-cytosine hydrogen bonding. However, little is known about the effect of CpG methylation on the i-motif. In this study, The CpG methylation-induced structural changes on the i-motif was examined by thermal stability, circular dichroism (CD) spectroscopy, and native-polyacrylamide gel electrophoresis (Native-PAGE) evaluation of five i-motif-forming DNAs from four cancer-related genes (*VEGF*, *C-KIT*, *BCL2*, and *HRAS*). This research shows that CpG methylation increased the transitional pH of several i-motif-forming DNAs and their thermal stability. When examining the effect of CpG methylation on the i-motif in the presence of opposite G4-forming DNAs, CpG methylation influenced the proportion of G4 and i-motif formation. This study showed that CpG methylation altered the stability and structure of the i-motif in CpG islands.

## 1. Introduction

In the nucleus, DNA is present as chromatin and assembles a complex known as the nucleosome [1]. Modifications of DNA and histones affect nucleosome formation and control gene expression [2]. Methylation of the fifth carbon of cytosine is an epigenetic modification that plays an important role in mammalian gene transcription and development [3]. Cytosine methylation occurs at CpG oligodeoxynucleotides (CpG-ODNs), which are subsequently recognized by the CpG methyl-CpG-binding domain protein family, resulting in the silencing of heterochromatin [4]. Up to 90% of all mammalian CpG ODNs are methylated and regulate gene expression [5]. Furthermore, CpG-ODNs are densely present in promoter regions known as CpG islands [5]. However, the methylation levels of CpG islands are low compared to that of CpG-ODNs outside the CpG islands [5]. Normal DNA methylation induces cell differentiation, whereas abnormal methylation is frequently recognized in CpG islands during severe diseases, such as cancer [6]. The G-rich sequences in CpG islands form G-quadruplexes (G4), which are formed by G-tetrads stacking and Hoogsteen bonds [7]. More than 300,000 G4-forming sequences are found in promoter regions [8] and bind to transcription factors to regulate gene expression [9,10]. Specificity protein 1 (Sp1) is a zinc finger transcription factor that binds to a variety of GC-rich DNA structures in the promoter region [11,12]. In our previous work, DNA methylation altered the structures of G4 in the promoter region of *VEGF* [13], *C-KIT*, *BCL2* [14], and *HRAS* [15,16] and the Sp1-binding ability [17]. Sp1 has many binding sites on CpG islands, whereas CpG methylation neither affects binding to CpG-ODN nor transcriptional suppression [18]. Thus, the regulatory role of G4 may be controlled via DNA methylation by altering the structure of G4.

The complementary strands of G4 have cytosine-rich sequences; therefore, the i-motif, which is known as a non-canonical DNA structure, can form in cytosine-rich sequences [19] by cytosine-cytosine hydrogen bonding [20]. Since this process requires hemi-protonated-cytosine, the i-motif is thought to form under acidic pH. Interestingly, under molecular crowding conditions [21] or in the presence of copper ions [22], the i-motif formation occurred at neutral pH. Recently, a genome-wide screening method for i-motif-forming DNA sequences under neutral pH has been reported [23]. Moreover, i-motif structure visualization using i-motif-binding antibodies in mammalian cells [24] and in-cell nuclear magnetic resonance has allowed for the evaluation of i-motif stability in living cells [25]. It has also been reported that cytosine methylation can increase the stability of the telomeric i-motif [26]. This suggests a relationship between CpG methylation and the i-motif while examining the effect of CpG methylation on single-stranded DNA. Here, since the structures of G4 in the promoter regions of *VEGF*, *C-KIT*, *BCL2*, *HRAS1*, and *HRAS2* are altered by CpG methylation, it is expected that the structures of the i-motif, and the complementary strands of these G4 sequences, are also altered by CpG methylation. We recently reported the stabilizing capability of CpG methylation on the i-motif present in the *VEGF* promoter region [27]. In this study, five i-motif forming DNAs from four cancer-related genes which form G4 structures are focused and the effects of CpG methylation on various i-motif structures of CpG islands were investigated via the evaluation of thermostability and i-motif structure. The circular dichroism (CD) spectra of i-motif-forming DNAs revealed that i-motif formation was pH-dependent. In addition, native-polyacrylamide gel electrophoresis (PAGE) suggested that an i-motif conformational change occurred following CpG methylation under the same pH conditions. Finally, a significant change in the CpG methylation trend was observed following CD spectra analyses of the i-motif in the presence of G4-forming complementary DNAs. This study will improve our understanding of the potential regulatory mechanisms of the i-motif and epigenetic phenomena.

## 2. Results and Discussion

Five different i-motif-forming DNAs from four cancer-related genes [17] were selected as candidates for examining the effects of epigenetic modifications (Table 1). The interaction between the i-motif and transcriptional activation for *BCL2* and *HRAS* i-motifs in CpG islands is described below. Considering that approximately 90% of CpG-ODNs are methylated in human cells [5], further studies are needed to characterize the effects of the CpG fully methylated i-motif on pH and thermal stability.

Circular Dichroism (CD) spectroscopy, a standard method for predicting the secondary structure of DNA, was used to analyze the epigenetic effect of CpG methylation on structural factors. Stabilization of the i-motif occurs under acidic conditions, indicated by the observed positive and negative peaks at 290 nm and 265 nm, respectively. These peaks are indicative of i-motif formation (Figure 1), and were consistent for the other four DNAs as well (Appendix A). The transitional pH indicates the equilibrium between random coil DNA and the i-motif and is generally recognized as an index for i-motif stability. Here, the transitional pH was calculated by fitting the molar ellipticities at 290 nm associated with pH change (Appendix A) [26], thereby indicating its range for unmethylated DNAs to be 6.02–6.81 (Table 1).

Contrarily, CpG methylated DNAs showed an increase in transitional pH values, particularly for the *C-KIT* and *HRAS2* i-motifs by 0.16 and 0.15, respectively (Table 1). CpG methylation had a small effect on the *VEGF* and *BCL2* i-motifs. However, CpG methylation altered the molar ellipticities of these DNAs at each pH, suggesting that structural differences were caused by the modification, irrespective of pH. An increase in transitional pH was observed for all DNAs, excluding the *HRAS1* i-motif. However, a unique slide at the negative peak of the *HRAS1* i-motif from 245 to 265 nm at pH 4.4 was observed (Appendix A), indicating a structural change subject to CpG methylation.

Native-polyacrylamide gel electrophoresis (PAGE) of i-motif DNAs was performed to investigate the effect of CpG methylation on these structures around their transitional pH. The ratio of formation of the i-motif structures was substantially changed by CpG methylation at transitional pH, \ as CpG methylation-induced structural changes seemed to be related to DNA mobility at this pH. In the absence of CpG methylation, *BCL2, HRAS1*, and *HRAS2* i-motifs showed smeared bands above the main band, suggesting the formation of multimers (Figure 2). However, no smeared bands were observed in the presence of CpG methylation. Additionally, CpG methylation shifted the band of the *HRAS2* i-motif lower than that of the unmethylated DNA. Equivalent trends were observed in the *BCL2* and *HRAS1* lanes.

Unmodified DNAs form unexpected structures and multimers, suggesting that CpG methylation can “fix” the structure and form a monomer-type i-motif. This phenomenon is likely caused by a structural alteration of the DNAs at transitional pH, which is unsuitable for the i-motif due to the relative paucity of hemi-protonated cytosine (Table 1). Previous studies have suggested that CpG methylation increases the formation of the i-motif in some DNA sequences [26,27,28]. Thus, CpG methylation may promote formation of the i-motif under the same pH conditions. Consistent with the native-PAGE results, the effect of CpG methylation on stability was caused by alterations in the DNA structure.

A previous study described the stabilization of the telomeric i-motif by CpG methylation, suggesting that methyl groups donate electrons to cytosine and gain their proximity to hemi-protonated cytosine [26]. To explore the effect of CpG methylation on the thermostability of i-motifs formed in CpG islands, thermal denaturation studies were performed by monitoring the CD spectrum at 288 nm while gradually increasing temperature from 10 °C to 95 °C around their transitional pH (Figure 3). The thermal denaturation curve of *HRAS2* demonstrated an elevated melting temperature (*T_m_*) from 31.5 °C to 36.0 °C at pH 6.8, indicating that CpG methylation contributed to its thermostability. The differences in the *T_m_* (Δ*T_m_*) of *VEGF*, *BCL2*, and *HRAS2* i-motifs were 2.9 °C, 1.9 °C, and 6.9 °C, respectively, at their transitional pH (Table 2). The *T_m_* of the *C-KIT* oligonucleotide was not determined since the CD values did not converge. These results indicated that CpG methylation stabilizes the i-motif under near-physiological conditions.

Recent studies have revealed that G4 and i-motifs are formed in human nuclei [24,29], and the number of these structures in the cell depends on the cell cycle. Although the number of G4 structures in the nuclei is maximized in the S phase, the i-motif formation frequency is at its peak in the G1/S phase. This indicates that G4 and i-motifs were formed reversibly. In addition, exclusiveness of the structures has been observed in some G4/i-motif-forming sequences [30]. However, in silico computational results have shown that the formation of its structure promotes the formation of the opposite structure [31]. These conflicting reports clearly indicate a gap in knowledge regarding the correlation between G4 and i-motif formation.

Based on this information, the stability of the i-motif in the presence of opposite G4-forming sequences was examined. *HRAS2* i-motif-forming DNA and *HRAS2* G4-forming DNA (*HRAS2* G4/i-motif mixed DNA) were mixed and the associated CD spectrum was obtained. Methylation in *HRAS2* i-motif-forming DNA significantly affects both thermal stability and pH stability. At pH 5.8, the *HRAS2* G4/i-motif-mixed DNA showed a broad positive peak at 260 and 290 nm and a negative peak at 245 nm, which differs from the typical CD spectrum associated with double-stranded DNA (Figure 4). However, in the presence of CpG methylation, the peak at 260 nm slightly decreased, whereas the peak at 290 nm was unaffected. These results suggest that *HRAS2* G4/i-motif-mixed DNA does not fully form duplexes and that the proportion of G4 and i-motif formation was changed; in particular, the proportion of G4-forming DNAs appeared to decrease slightly. DNA native- PAGE of *HRAS2* G4/i-motif-mixed DNA at pH 5.8 (Appendix A) was also performed. Unmethylated and methylated G4/i-motif-mixed DNA showed distinct band patterns that are indicative of the presence of several structures and complexes apart from the duplex. This demonstrates that CpG methylation altered the structure of G4/i-motif-mixed DNAs. Given that the CpG methylation increased the stability of the i-motif, CpG methylation not only affected the i-motif-forming single-stranded DNA, but also the i-motif/G4 complex in the presence of the opposite G4-forming DNAs.

## 3. Conclusions

These results indicate that CpG methylation changed the formation tendency of the i-motif and its thermostability at different pH levels with several oligoDNAs bearing different sequences. Native-PAGE analysis was performed to investigate the CpG methylation-induced conformational changes of the i-motif. Furthermore, CpG methylation affected the structure of the i-motif in the presence of G4-forming DNAs. In particular, the *HRAS2* G4/i-motif mixture did not appear to fully form a duplex at pH 5.8.

It was shown that CpG methylation of the i-motif in the promoter region of several cancer-related genes affected the formation tendency of the i-motif with respect to pH and thermostability. CpG methylation also induced i-motif structural changes in the presence of G4-forming DNAs, which are complementary strands of the i-motif, assuming duplex formation. This indicates the involvement of CpG methylation during spontaneous i-motif and G4 formation in the promoter regions, with duplex unfolding related to transcriptional control. This research contributes to the elucidation of the transcriptional regulation mechanism controlled by CpG methylation corresponding to the i-motif structure in the promoter region.

## 4. Materials and Methods

### 4.1. Oligonucleotide Preparation

All oligonucleotides used in this study were synthesized by FASMAC Co., Ltd. (Kanagawa, Japan) and Eurofins Genomics, K.K. (Tokyo, Japan). The DNA sequences used in the present study are listed in Appendix A. Oligonucleotides were diluted in 10 mM each of Tris-HCl, NaCl (pH 8.0, 7.4, 7.0), MES, NaCl (pH 6.8, 6.6, 6.4, 6.2, 6.0, 5.8), or sodium acetate buffer (CH_3_COOH or CH_3_COONa; pH 5.1, 4.4). Oligonucleotides were folded at 95 °C for 10 min and then gradually cooled to 25 °C.

### 4.2. Circular Dichroism Spectrum Measurement and Melting Temperature (Tm)

Circular Dichroism (CD) spectra of the oligonucleotides were measured at 20 °C at a wavelength range of 220–320 nm using a J-720 or J-820 spectropolarimeter (JASCO, Tokyo, Japan) equipped with a 1-cm path cuvette. DNA aliquots were diluted in the buffer to a final concentration of 5.0 µM and folded by thermal processing as described above. The melting curve was recorded at approximately 288 nm during cuvette heating from 10 °C to 95 °C.

### 4.3. Data Analysis

The transitional pH was calculated by fitting the molar ellipticities at 290 nm of each pH, as described above. The molar ellipticities at pH 4.4 and 8.0 were set as 100% and 0%, respectively, and the normalized molar ellipticities were calculated. The curve was fitted using Image J software (Wayne Rasband (NIH), MD, USA, ImageJ bundled with 64-bit Java 1.8.0_172). The transitional pH was determined as the inflection point pH of the fitting curve.

The melting temperature (*T_m_*) was calculated by fitting the molar ellipticities at 288 nm at transitional pH of each DNAs. The molar ellipticities at 10 °C and 95 °C were set as 100% and 0%, respectively, and the normalized molar ellipticities were calculated. The curve was fitted using Image J software (Wayne Rasband (NIH), MD, USA, ImageJ bundled with 64-bit Java 1.8.0_172). The melting temperature was determined as the inflection point temperature of the fitting curve.

### 4.4. DNA Native-Polyacrylamide Gel Electrophoresis (Native-PAGE)

DNA native-polyacrylamide gel electrophoresis (Native-PAGE) was performed on 20% polyacrylamide gel prepared using TAE buffer, while the pH was adjusted for each experiment. The oligonucleotides were heat-treated as described above before undergoing electrophoresis at 100 V for 120 min at 4 °C. The gel was stained with StainsAll (Sigma-Aldrich, St. Louis, MO, USA), and images were obtained using a Typhoon8600 (GE Healthcare, Little Chalfont, UK).

## Figures and Tables

**Figure 1 ijms-23-06467-f001:**
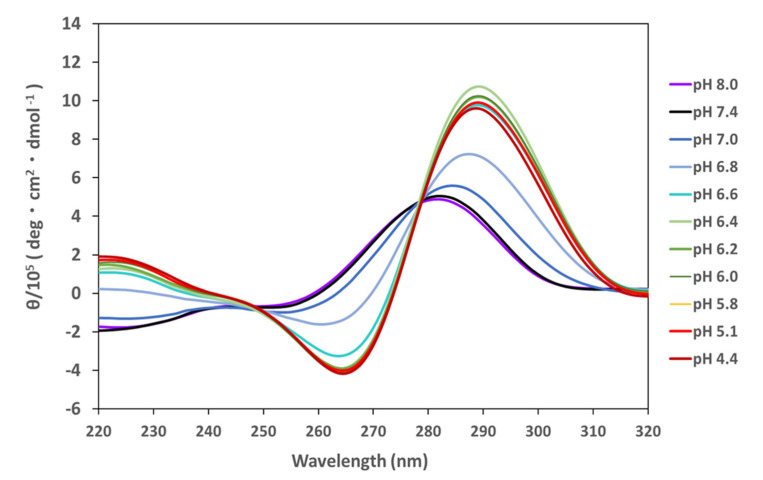
CD spectrum of *HRAS2* i-motif in 10 mM Tris-HCl, 10 mM NaCl (pH 8.0, 7.4, 7.0), 10 mM MES, 10 mM NaCl (pH 6.8, 6.6, 6.4, 6.2, 6.0, 5.8), or 10 mM sodium acetate buffer (10 mM CH_3_COOH or CH_3_COONa, pH 5.1, 4.4) at 20 °C.

**Figure 2 ijms-23-06467-f002:**
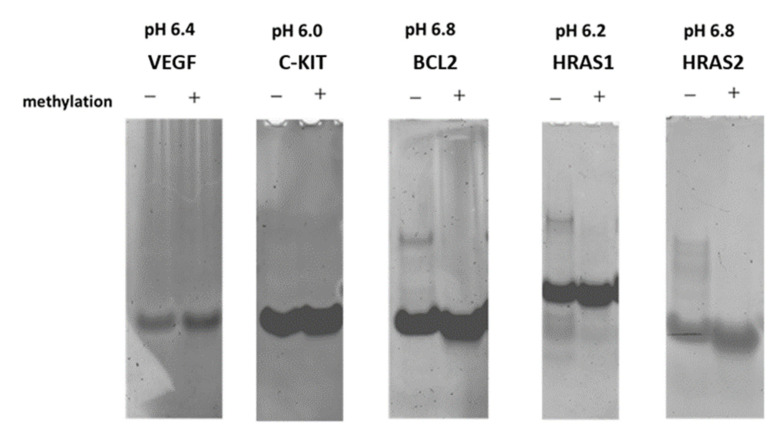
Native-PAGE of the DNAs at each transitional pH of unmethylated i-motifs. The presence (+) and absence (−) of CpG methylation (+) is indicated at the top of each lane.

**Figure 3 ijms-23-06467-f003:**
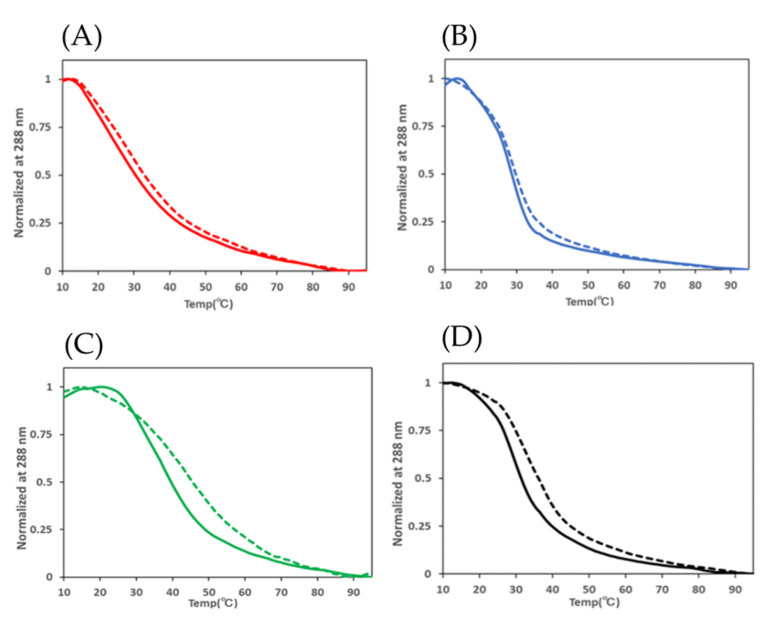
Evaluation of *T_m_* for (**A**) *VEGF* i-motif, (**B**) *BCL-2* i-motif, (**C**) *HRAS1* i-motif, and (**D**) *HRAS2* i-motif. Solid lines indicate unmethylated samples, and dashed lines indicate methylated samples.

**Figure 4 ijms-23-06467-f004:**
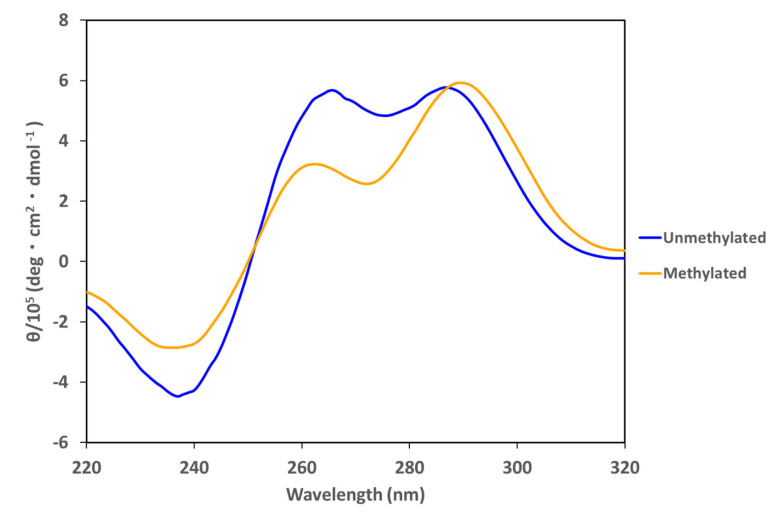
CD spectrum of *HRAS2* G4/i-motif-mixed DNAs at pH 5.8 of unmethylated (blue) and methylated (orange) DNAs.

**Table 1 ijms-23-06467-t001:** Transitional pH of unmethylated and methylated *VEGF*, *C-KIT*, *BCL-2*, *HRAS1,* and *HRAS2* i-motifs. CpG methylation increased the transitional pH of several i-motif-forming DNAs.

i-Motif	Transitional pH Unmethylated	Transitional pH Methylated	ΔTransitional pH
*VEGF*	6.47	6.50	0.03
*C-KIT*	6.02	6.18	0.16
*BCL2*	6.81	6.82	0.01
*HRAS1*	6.27	5.87	−0.4
*HRAS2*	6.80	6.95	0.15

**Table 2 ijms-23-06467-t002:** *T_m_* of unmethylated and methylated *VEGF*, *BCL-2*, *HRAS1*, and *HRAS2* i-motif. CpG methylation increased *T_m_* values of all i-motif forming DNAs.

i-Motif	Transitional pH	*T_m_* (°C) Unmethylated	*T_m_* (°C) Methylated	Δ*T_m_*
*VEGF*	6.4	29.6	32.7	2.9
*BCL2*	6.8	28.1	30.0	1.9
*HRAS1*	6.2	39.6	46.5	6.9
*HRAS2*	6.8	31.5	36.0	4.5

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
