# Peer review of "CpG Methylation Altered the Stability and Structure of the i-Motifs Located in the CpG Islands"

_ijms, 2022, doi:10.3390/ijms23126467_

Round 1

Reviewer 1 Report

This is a well-prepared manuscript deals with an important topic. However, some concerns need to be addressed to fit for publication as follows:

1.      It is not preferred to begin sentences with abbreviations like CPG in line 11. Please revise the whole manuscript for such an error.

2.      The manuscript needs to be revised for the English and the overall writing style. The writing style should be formal from the third-person perspective. Do not use we or our.

3.      More keywords should be added.

4.  In the methods section the authors should add more information on the software used in the analysis of the data.

5.      In the conclusion section: this sentence “Some reports have suggested a relationship between CpG methylation and the 193 i-motif while examining the effect of CpG methylation on single-stranded DNA [26]” should be transferred to the introduction with more details of these reports.

Author Response

Dear Prof. Dr. Caitlin Huang and reviewers:

We would like to thank the reviewers for their helpful comments. We would like to submit our revised manuscript, ijms-1762667, to International Journal of Molecular Sciences. We have fully revised our manuscript in accordance with the suggestions and provided point-to-point responses to the reviewer’s comments. The revised text in the manuscript is represented in red.

Reviewer 1

  1. It is not preferred to begin sentences with abbreviations like CPG in line 11. Please revise the whole manuscript for such an error.

Response

        Thank you for your kind advice. We have revised the abbreviations used at the beginning of sentences in the manuscript (including line 11).

  1. The manuscript needs to be revised for the English and the overall writing style. The writing style should be formal from the third-person perspective. Do not use we or our.

Response

Thank you very much for your comment. We revised the writing style of the manuscript. A certificate of English proofreading by professionals (Editage) is attached.

  1. More keywords should be added.

Response

Thank you very much for the valuable comment. We added some keywords “epigenetics”, “pH dependence”, “structural change”, “thermal stability”. In the current study, we showed that CpG methylation altered the pH dependence and thermal stability of the i-motif structure, so we considered it appropriate to add these keywords.

  1. In the methods section the authors should add more information on the software used in the analysis of the data.

Response

We added a software analysis section (lines 232–243) and explained how to obtain the fitted curves and determine the transitional pH and melting temperature. In addition, we added the fitted curves of the transitional pH to the supporting information.

  1. In the conclusion section: this sentence “Some reports have suggested a relationship between CpG methylation and the 193 i-motif while examining the effect of CpG methylation on single-stranded DNA [26]” should be transferred to the introduction with more details of these reports.

Response

We appreciate the reviewer’s helpful comment. We have moved this sentence to the Introduction (line 60).

We believe we addressed all the concerns raised by the reviewer and revised those parts accordingly. We really appreciate the reviewers for their helpful comments, and your kind consideration of our manuscript for publication in International Journal of Molecular Sciences.

We look forward to hearing good news.

Yours sincerely,

Kazunori Ikebukuro, Ph.D. (Engineering)

Reviewer 2 Report

Comments:

1. On Figure 2, why BCL2, C-KIT, and VEGF have only one pH study?

2. Authors' previous study used the same 4 genes. Need to expand to study more --motif-forming genes.

3. How about tumor suppressor genes which also have i-motif-forming DNAs. The current study focuses on oncogenes.

4. How about other i-motif-forming genes such as DAP, RAP17, SMARCA4, ACC1, Rb, SERT, c-MYC?

Author Response

Dear Prof. Dr. Caitlin Huang and reviewers:

We would like to thank the reviewers for their helpful comments. We would like to submit our revised manuscript, ijms-1762667, to International Journal of Molecular Sciences. We have fully revised our manuscript in accordance with the suggestions and provided point-to-point responses to the reviewer’s comments. The revised text in the manuscript is represented in red.

Reviewer 2

  1. On Figure 2, why BCL2, C-KIT, and VEGF have only one pH study?

Response1

Thank you for your question. The ratio of formation of i-motif structures was substantially changed by CpG methylation at the transitional pH. Therefore, CpG methylation-induced structural changes were investigated by DNA mobility using native polyacrylamide gel electrophoresis (PAGE) at each transitional pH. To explain the change in transitional pH by CpG methylation in detail, we added information on how to obtain fitted curves and determine transitional pH in the supporting information (Figure S6-S10). Moreover, we added an explanation of the importance of performing native-PAGE in the main manuscript (including lines 103-107).

  1. Authors' previous study used the same 4 genes. Need to expand to study more --motif-forming genes.

Response2

Thank you very much for your helpful comment. We previously reported that the sequences in the promoter region of VEGF, C-KIT, BCL2, HRAS formed G-quadruplexes (G4), and the structures of G4 were altered by CpG methylation. Therefore, it was expected that the structure of the i-motif, which might be formed by the complementary strands of these G4 sequences, would also be altered by CpG methylation. We have added an explanation of the relationship between G4 and the i-motif in lines 60-65 in the text because it is important. In addition, we added sentences of why we focused on these four cancer-related genes in lines 61-64.

  1. How about tumor suppressor genes which also have i-motif-forming DNAs. The current study focuses on oncogenes.
  2. How about other i-motif-forming genes such as DAP, RAP17, SMARCA4, ACC1, Rb, SERT, c-MYC?

Response 3 and 4

We appreciate the reviewer’s valuable suggestions. It is necessary to evaluate other i-motif-forming genes. However, as described above, the current study aimed to evaluate the complementary sequences of G4 from cancer-related genes, whose structures were altered by CpG methylation. Therefore, we considered it sufficient to evaluate the i-motif-forming sequences in this study. These suggested genes will be investigated in future studies.

We believe we addressed all the concerns raised by the reviewer and revised those parts accordingly. We really appreciate the reviewers for their helpful comments, and your kind consideration of our manuscript for publication in International Journal of Molecular Sciences.

We look forward to hearing good news.

Yours sincerely,

Kazunori Ikebukuro, Ph.D. (Engineering)

Round 2

Reviewer 2 Report

No more comments